# Impact of COVID-19-Related Social Isolation on Behavioral Outcomes in Young Adults Residing in Northern Italy

**DOI:** 10.3390/ijerph192416496

**Published:** 2022-12-08

**Authors:** Alessandra Patrono, Azzurra Invernizzi, Donatella Placidi, Giuseppa Cagna, Stefano Calza, Manuela Oppini, Elza Rechtman, Demetrios M. Papazaharias, Abraham Reichenberg, Roberto G. Lucchini, Maurizio Memo, Elisa Ongaro, Matteo Rota, Robert O. Wright, Stefano Renzetti, Megan K. Horton

**Affiliations:** 1Department of Molecular and Translational Medicine, University of Brescia, 25121 Brescia, Italy; 2Department of Environmental Medicine and Public Health, Icahn School of Medicine at Mount Sinai, New York, NY 10029, USA; 3Department of Medical and Surgical Specialties, Radiological Sciences and Public Health, University of Brescia, 25121 Brescia, Italy; 4Department of Psychiatry, Icahn School of Medicine at Mount Sinai, New York, NY 10029, USA; 5Department of Environmental Health Sciences, Robert Stempel College of Public Health and Social Work, Florida International University, Miami, FL 33199, USA

**Keywords:** social isolation, mental health, COVID-19, young adults

## Abstract

Social isolation affects our emotions, behavior and interactions. Worldwide, individuals experienced prolonged periods of isolation during the first wave of the COVID-19 pandemic when authorities-imposed restrictions to reduce the spread of the virus. In this study, we investigated the effects of social isolation on emotional and behavioral outcomes in young adults from Lombardy, Italy, a global hotspot of COVID-19. We leveraged baseline (pre-social isolation) and follow-up (mid- or post-isolation) data collected from young adults enrolled in the ongoing, longitudinal Public Health Impact of Metals Exposure (PHIME) study. At baseline, 167 participants completed the ASEBA questionnaires (ASR/YSR) by web link or in person; 65 completed the ASR 12–18 weeks after the onset of restrictions. Using the sign test and multiple linear regression models, we examined differences in ASR scores between baseline and follow-up adjusting for sex, age, pre-pandemic IQ and time with social restrictions (weeks). Further, we examined interactions between sex and time in social isolation. Participants completed the ASR after spending an average of 14 weeks in social isolation (range 12–18 weeks). Thought problems increased between baseline and follow-up (median difference 1.0; 1st, 3rd quartile: −1.0, 4.0; *p* = 0.049). Among males, a longer time in social isolation (≥14 weeks) was associated with increased rule-breaking behaviors of 2.8 points. These results suggest the social isolation related to COVID-19 adversely impacted mental health. In particular, males seem to externalize their condition. These findings might help future interventions and treatment to minimize the consequences of social isolation experience in young adults.

## 1. Introduction

The COVID-19 pandemic has led to unprecedented social distancing behavior to limit the spread of the virus [1,2]. These measures impacted the general population, not only those found to be infected or exposed to the disease. Social distancing measures can effectively counter the spread of the disease [3], but can have an unprecedented impact on mental health and psychological well-being.

In Italy, the Lombardy area was the epicenter of the infection and one of the first places in the Western world confronted with COVID-19. No medications or vaccinations were available during the first wave of COVID-19. Therefore, the Italian government implemented a non-pharmacological measure referred to as lockdown, a forced and prolonged period of social restrictions, distancing and isolation [4,5] to stem the spread of infection. Lockdown social isolation refers to the “inadequate quality and quantity of social relationships with other people at the individual, group, community level and the wider social environment in which human interaction occurs” [6]. Italy was the first country to enter a COVID-19-related lockdown [7]. The lockdown and social isolation began in northern Italy on 23 February 2020 [8]. Increasingly restrictive decrees followed gradually up to 9 March 2020, and the restraining measures were extended throughout Italy from 11 March 2020 [9]. In Italy, during this period, only essential activities and shops were accessible (i.e., medical services, grocery stores), individuals were allowed to leave their homes only for demonstrated needs, such as for health reasons, shopping for basic needs and for work (if it was not possible to work from home) [7]. Social gatherings were minimized or prohibited [10]. The restriction measures in Italy have gradually decreased starting with a more extensive opening of shops, the permission to leave one’s home to reach relatives and the possibility of being able to attend social events and equipped with a mask. On 11 June, the containment measures were eased again but conditions of distancing, the use of masks and the discouragement of social situations were maintained. Our investigation took place when participants were subjected to socially restrictive conditions and had recently been subjected to social isolation.

Although these restrictive measures successfully prevented more serious consequences of the COVID-19 pandemic, the social isolation may have resulted in mental health conditions [4]. Extended social isolation conditions related to COVID-19 have been associated with short- and long-term psychosocial and mental health consequences among all ages of the population [11]. The magnitude of the impact is influenced by many risk factors such as gender [12], age [13], economic disadvantage [14] and pre-existing health conditions [15]. In general, sex (female), age (individuals 18–30 years and over 60 years of age) and education (higher education) were associated with the highest levels of mental health problems following COVID-19-related social isolation [16,17] such as anxiety, sleep disorders and depression [16].

Although several studies have focused on mental health assessment in different subgroups of the population and especially investigated the effect of social restrictions in the elderly [10], there are few studies that have longitudinal data (baseline and follow-up) of the impact of COVID-19-related social isolation on the mental health of healthy young adults. In this study, we examine the impact of COVID-19-related social isolation on emotional and behavioral outcomes among healthy young adults living in northern Italy (Province of Brescia), one of the first global hotspots of COVID-19. Using information on behavioral outcomes collected prior to and following participants’ experience of social isolation, we aim to quantify the impact of social isolation on young adult mental health to inform future interventions to minimize or eliminate the consequences of social isolation experience in healthy young adults.

## 2. Materials and Methods

### 2.1. Participants

Participants were part of the Public Health Impact of Metals Exposure (PHIME) study, an ongoing longitudinal cohort study of adolescents in the Province of Brescia, northern Italy. PHIME was designed to assess cognitive and behavioral function in adolescents and young adults with environmental exposure to neurotoxic metals. Participants were never to have received a psychological or neuropsychological diagnosis. Other enrollment, inclusion and exclusion criteria for the PHIME study are described in detail elsewhere [18,19,20,21]. Upon enrollment, PHIME participants participated in a baseline in-person visit consisting of self- and interviewer-assisted questionnaires capturing sociodemographic characteristics (i.e., sex, date of birth, residential address, parental education and occupation) and neurodevelopmental outcomes including the Kaufman Brief Intelligence Test, second edition (K-BIT 2) [22] for IQ and the Achenbach System of Empirically Based Assessment (ASEBA) Youth Self Report (YSR) [23] or (ASEBA) Adult Self Report (ASR) [24] for behavioral and emotional regulation (Section 2.3). As part of the PHIME study, 167 participants (ages 19.3 years ± 2.3) completed the YSR or the ASR in person with a trained psychologist prior to the beginning of COVID-19-related social isolation (on average, the first visit was performed 81.8 ± 43.2 weeks prior to the first day of social isolation). To assess the impact of social isolation on emotional and behavioral outcomes, we re-administered the ASR via an online platform (REDCap^®^, Research Electronic Data Capture, Vanderbilt University, Nashville, TN, USA) 12–18 weeks following the onset of social isolation measures (on average, the subjects answered the questionnaire after 13.4 ± 1.4 weeks). We distributed the web link to all 167 PHIME participants who completed the baseline assessment; 40% (65/167) of participants completed the online ASR. During the second time point, no information relating to the SES and the IQ was collected again, as these variables were considered stable over a short time after the first administration.

Eligible participants received a detailed description of the study procedures before consenting to participate. The parents of the minors during the baseline phase received an informed consent form to be signed. The Institutional Review Boards of the Ethical Committee of Brescia, the Icahn School of Medicine at Mount Sinai and the University of California, Santa Cruz approved all PHIME study protocols.

### 2.2. ASEBA Young and Adult Self Report (YSR and ASR) Questionnaires

The Achenbach System of Empirically Based Assessment (ASEBA) offers a comprehensive approach to assessing adaptive and maladaptive functioning. During the baseline PHIME visit, adolescent participants (ages 15–17 years; *n* = 45) completed the Youth Self Report (YSR) questionnaire [23,25] and adult participants (ages 18–25 years; *n* = 122) completed the Adult Self Report (ASR) questionnaire [26]. The YSR questionnaire is designed for self-reporting in the 11–17 years age range; the ASR is appropriate for adults ages 18–59 years. ASR questionnaires evaluate the following clinical areas: (I) Anxious/Depressed; (II) Withdrawn; (III) Somatic Complaints; (IV) Thought Problems; (V) Attention Problems; (VI) Aggressive Behavior; (VII) Rule-Breaking Behavior; (VIII) Intrusive. An Internalization Problem Composite Scale is aggregated from the individual symptoms scales: Anxiety (18 items), Withdrawn (9 items) and Somatic Complaints (12 items). An Externalizing Problem Composite Scale is composed of: Aggressive Behavior (15 items), Rule-Breaking Behavior (14 items) and Intrusive Behavior (6 items). The other scales concern Attention Problems (15 items) and Thought Problems (10 items). The scale Other Problems (21 elements) includes elements that do not frame any syndrome. The remaining 11 items measure adaptive functioning. For the following study, we used the version validated on the Italian population.

The clinical scales investigated by the ASR are comparable to those of the YRS with some changes related to the adaptation of the items by age. In the YSR version, the component of Depression is investigated both by scale I and II; the ASR scale VIII Intrusive corresponds to the YSR scale V, Thought Problems.

Since the seven ASR syndromes have YSR-rated counterparts, questionnaire scores can be directly compared [27]. A score from 0 (behavior/problem absent) to 2 (behavior/problem present) is applied to each item that makes up the individual scales considered in the YSR/ASR questionnaires. Each scale will then assume a numerical value which is transformed into a T score in a range from 50 to 100. Scores between 50 and 64 are considered normal; scores between 65 and 70 are considered borderline; scores above 70 are considered clinically significant.

Both the YSR and ASR yield a Total Problems score indicating the overall psychopathological assessment of the individual (a higher score indicates greater psychopathology). Symptomatic scales correlate with the DSM-oriented diagnosis (i.e., ASR depressive symptoms correlate with DSM-diagnosed depression).

### 2.3. Covariate Data

Sociodemographic data (i.e., participant sex and age, and parental occupation and education) were collected at the baseline assessment through questionnaires. Intelligence quotient (IQ) was measured using the Kaufman Brief Intelligence Test, 2nd edition (KBIT-2) [28], a short measure of verbal and non-verbal intelligence for children, adolescents and adults, aged 4 to 90 years. The verbal and non-verbal scores yield a composite IQ score that can be considered as a measure of general intelligence with good correlations with other tests of intellectual functioning [28]. An index of family socioeconomic status (SES; low, medium or high) was calculated from parental age, occupation and education [29].

### 2.4. Statistical Analysis

Descriptive statistics were used to assess the distribution of variables; continuous variables are analyzed using the median and the first and third quartiles because of the skewed distribution and, for categorical variables, we used absolute frequencies and percentages. Student’s *t*-tests with Welch’s correction for continuous variables and chi-squared (χ^2^) tests for categorical variables were used to examine differences in demographic characteristics across the participants. The time spent in social isolation was calculated in weeks starting from the start of social isolation (9 March 2020) and the date the participants completed the follow-up ASR online. Time spent in isolation was calculated based on the median (14 weeks); low = less than 14 weeks between the initiation of the first restrictions and the follow-up questionnaire response, high = greater than or equal to 14 weeks between onset of social isolation and follow-up questionnaire response. Figure 1 describes the distribution of time spent in social isolation starting from the first day of social isolation to the last day when a response to the questionnaire was collected. Most of the data were collected around 12 weeks from the start of the social isolation (*n* = 24, 36.9%) while the last responses were received after around 18 weeks (*n* = 2, 3.1%).

We used the sign test, a non-parametric test to assess differences between paired observations, to assess the difference between YSR/ASR symptoms before and after social isolation. The choice of the sign test was also driven by the non-symmetric distribution of differences. We then examined how the amount of time in social isolation, defined as the time elapsed between the first day of social restrictions and the follow-up visit, impacted differences in YSR/ASR scores. We applied a linear regression model to examine how time elapsed in isolation (independent variable) predicted the change in YSR/ASR scores, adjusting for age, sex, baseline SES and IQ. We then determined whether the associations between time in social isolation and the change in ASR scores differed by sex through a multiplicative interaction term. Statistical significance level was set at 5% for all tests. All the statistical analyses were performed with R (version 4.1.0).

## 3. Results

### 3.1. Sociodemographic Characteristics

Sociodemographic characteristics of PHIME participants included in this study are presented in Table 1. In total, 65 participants (26 male, 19.8 +/− 2.4 years) repeated the ASR during or following the social isolation (i.e., follow-up). Participants experienced an average of 14.6 +/− 9.5 weeks with social restrictions before completing the follow-up ASR questionnaire (Figure 1). The average IQ was 106.1 (SD 9.7). No YSR/ASR scores indicated problematic behaviors at baseline or follow-up (Appendix A). Sociodemographic characteristics and baseline ASR scores of those participants who completed the online follow-up assessment did not differ from those who did not complete the assessment. The amount of time (weeks) spent in social isolation ranged from 12–18 weeks, the average time spent in isolation was 14 weeks (Figure 1).

### 3.2. Social Isolation and Behavioral Outcomes

We observed no differences in Total Problems reported at baseline and follow-up (Figure 2A). Participants reported significantly more Thought Problems at follow-up (sign test; 51.5 (50.0, 55.8) vs. 53.5 (51.0, 58.0), *p* = 0.049; Figure 2B). None of the other ASR scales differed significantly between baseline and follow-up (Appendix A).

### 3.3. Length of Social Isolation and Behavioral Outcomes

Though not significant, we observed a trend between spending a longer amount of time in social isolation (<14 compared to ≥14 weeks) and an increase of 1.73 points in Rule-Breaking Behaviors was found (linear regression, *β* = 1.73; 95% confidence interval (CI): −0.03, −3.48, *p* = 0.053).

We did not observe significant differences between baseline and follow-up in the other YSR/ASR symptom scales or internalizing/externalizing composite scales (Appendix A).

### 3.4. Sex-Specific Effects of Social Isolation on Behavioral Outcomes

In the interaction between time and sex analysis (Appendix A), the amount of time spent in social isolation was significantly associated with increased Rule-Breaking Behavior in males only (i.e., average change in Rule-Breaking Behavior among males with a higher social isolation time = 2.8, 95%CI 0.06, 5.5, *p* = 0.046; Figure 3). Male participants who spent more time in social isolation (≥14 weeks) reported a 3-point increase in Rule-Breaking Behavior compared to males who spent less time (<14 weeks) in social isolation. No differences in the association between the time elapsed in social isolation and ASR scores were found for female participants.

### 3.5. Results of the Other Clinical Scales

As shown in Appendix A, the other clinical scales investigated did not show significant differences between baseline and follow-up. The total scale on externalized problems and internalizing problems is also stable, demonstrating how the impact of social isolation has been specific and circumscribed.

## 4. Discussion

In this study, we assess the impact of COVID-19-related social isolation on baseline (pre-isolation) and follow-up (mid- or post-isolation) behavioral outcomes in young adults enrolled in an ongoing longitudinal PHIME cohort study in northern Italy. Our findings suggest that social isolation is associated with increased Thought Problems. Further, the length of time spent in social isolation more adversely impacts males compared to females; males who spend more time in social isolation reported more Rule-Breaking Behavior.

In our study, males reported higher Rule-Breaking Behavior scores after social isolation than females. Further, spending a longer amount of time in social isolation increased the severity of Rule-Breaking Behavior scores (i.e., more time, more rule breaking). The construct of Rule-Breaking Behavior is defined as “non-compliance with the applicable regulatory expectations of the group” [30] and is related to disinhibition [31]. The general construct of rule breaking is considered a transitory factor within the behavior and mediated by the environmental situation [32,33]. Our findings in males contribute to the literature on social isolation and behavioral outcomes as most of the previous studies focus on female mood disorders related to the pandemic and social isolation [34]. Our data are fairly consistent with studies that broadly analyze gender differences in typical traits in mental disorders, with a higher frequency of behavioral outcomes in males [35] although these series may have been influenced by bias [36,37].

The worsening of these clinical scales, with regard to the social isolation period, is theoretically and clinically significant and reflects the need to implement intervention dynamics aimed at containing or preventing long-term effects. Creating free and easily accessible support networks for young adults is a solution that should be promoted. These networks, which can also be created online, could be facilitated by general practitioners and psychologists at the local level in the places most frequented by young adults, including university institutions. In particular, in view of the delivery of online therapies, the approaches of cognitive behavioral therapy, dialectical behavioral therapy and mind–body practice techniques have emerged as valid strategies to counteract the emerging symptomatology [38]. Thinking problems and anxiety levels may have been supported by the growing phenomenon of cyberchondria: a behavior characteristic of an excessive online search for medical information associated with rising levels of health anxiety [39]. Furthermore, previous research has found that receiving health information from the internet was associated with poorer psychological well-being [40]. This may become particularly true in a pandemic era, leading the World Health Organization to speak of two major threats to public health: the pandemic and the infodemic [31].

Our unique study design and population, located in one of the first global hotspots of the first wave of the COVID-19 pandemic, provided the opportunity to examine the impact of the pandemic-related social isolation on healthy young adults. Our findings suggesting that males may be more vulnerable to the impacts of social isolation on Rule-Breaking Behavior could help direct targeted interventions. In general, males are less likely to seek therapeutic interventions to treat mental health or mood-related disorders [41]. Based on the externalizing symptomatology that drives male behavior, initiatives that focus more on attention to functioning than on emotionality should be considered. General practitioners must be instructed to differentiate gender-specific alarm bells for subsequent referral to specialist treatment.

## 5. Limitations

The sample is relatively small and the participation rate was modest (40%) but we have a unique sample: healthy young adults living in one of the main COVID-19 disease hotspots in Europe at the beginning of the pandemic. We were able to collect the information during two time points in a period of time sufficient to evaluate behavioral changes due to social isolation. The low compliance could have driven a selection bias, with a possible tendency toward responses to the questionnaire only by the most emotionally affected subjects with a tendency of the most affected subjects to be interested in participating in the survey. However, although the sample is not very large, it assumes importance due to the possibility of being able to compare the scores with the previous administration of the questionnaire. Another limitation is the lack of information on COVID-19 infection and its possible impact of emotional and behavioral outcomes. At the time of our follow-up, accessibility of antigen and antibody verification of the presence of the disease was limited and took place only in the presence of symptoms. Rapid testing was not widespread. The socioeconomic data were not collected in the two time points, only in the first time point. Future investigations could investigate this aspect further.

## 6. Conclusions

To conclude, this study demonstrates how COVID-19 social restriction policies negatively impacted on mental and behavioral health in healthy young adults. The worsening of clinical scales of ASR, with regard to the pandemic period, is theoretically and clinically significant and reflects the need to implement intervention dynamics aimed at containing or preventing long-term effects of social isolation. Future studies are needed to understand how targeted interventions, based on the results of this and other similar research, can address changes in public health well-being.

## Figures and Tables

**Figure 1 ijerph-19-16496-f001:**
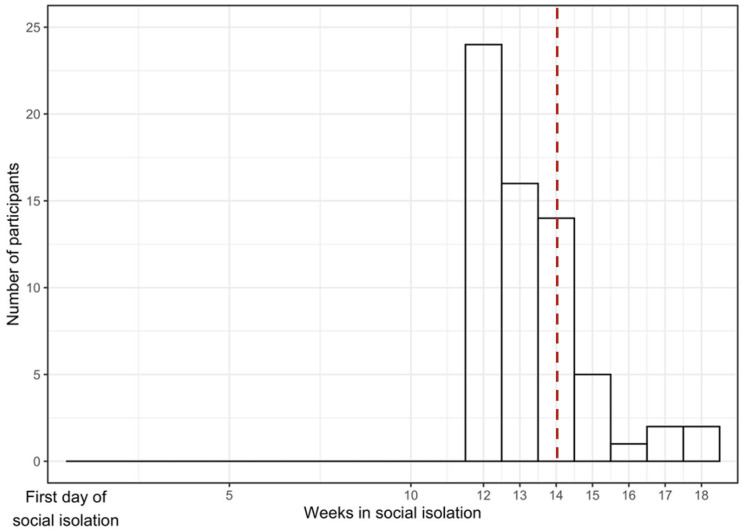
Time elapsed in social isolation. In total, 65 participants answered the questionnaire (axis *y*). The amount of time spent in isolation (axis *x*) was calculated as the number of weeks elapsed between the start of social isolation (3 September 2020) and the administration of the follow-up ASR. The red dashed line indicates the median (14 weeks). We categorized this variable as low (<14 weeks in social isolation) and high (≥14 weeks in social isolation) based on the median weeks spent in social isolation prior to assessment.

**Figure 2 ijerph-19-16496-f002:**
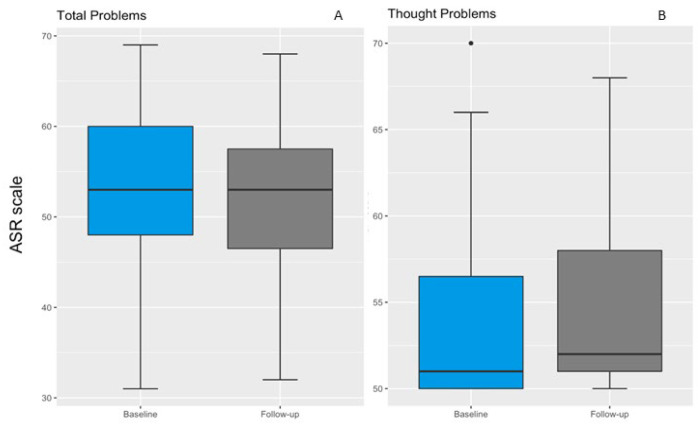
Differences in ASR score between baseline and follow-up. The boxplot shows YSR/ASR scores for Total Problems (**A**) and Thought Problems (**B**) at baseline (blue box) and follow-up (gray box). The error bars are the 95% confidence interval, the bottom and top of the box are the 25th and 75th percentiles, the line inside the box is the 50th percentile (median), and any outliers are shown as open circles. No differences are shown in Total Problems score scales between baseline and follow-up. Thought Problems are significantly higher in the follow-up assessment, with an overall clinical worsening in the post-social isolation period. The sign test was applied to test the difference between baseline and follow-up.

**Figure 3 ijerph-19-16496-f003:**
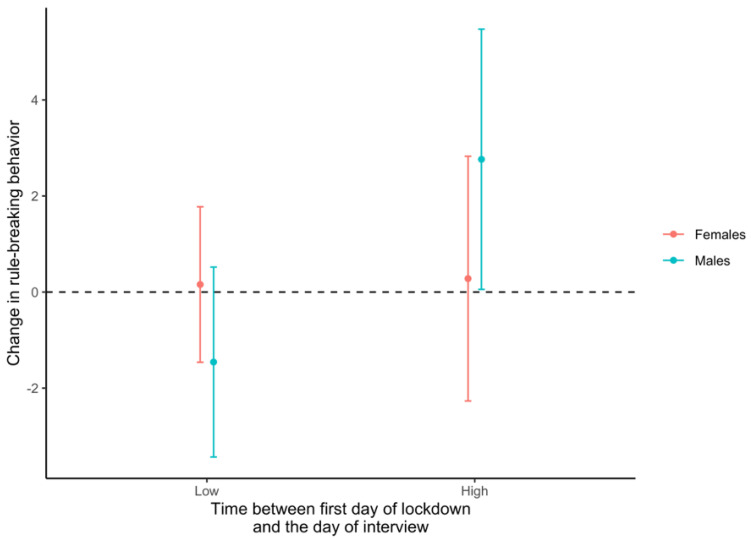
Sex-specific effects of social isolation and Rule-Breaking Behavior. Results from the linear regression model including an interaction term between time in social isolation and sex. Here are displayed the marginal effects of the time spent in social isolation (<14 weeks vs. ≥14 weeks vs. low) on the difference in the ASR Rule-Breaking Behavior score by sex. The model was adjusted by age (years), SES and IQ. The statistical significance for males with a longer time spent in social isolation is *p* = 0.046.

**Table 1 ijerph-19-16496-t001:** Sociodemographic characteristics of PHIME participants included in this study at baseline and follow-up (*n* = 65).

Characteristics	Baseline (*n* = 65)
**Age (years)**	
mean ± sd	19.8 (2.4)
**Sex (*n*, %)**	
Male	26 (40%)
Female	39 (60%)
**Socioeconomic status (*n*, %)**	
Low	
Medium	17 (26.2%)
High	32 (49.2%)
**IQ**	16 (24.6%)
Mean ± sd	106.1 (9.7)

Note: Mean, standard deviation (sd), range (minimum and maximum values) and percentage (%) are reported.

## Data Availability

Not applicable.

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
