# Peer review of "Impact of COVID-19-Related Social Isolation on Behavioral Outcomes in Young Adults Residing in Northern Italy"

_ijerph, 2022, doi:10.3390/ijerph192416496_

Round 1
Reviewer 1 Report
See attachment

Reviewer 2 Report
The submission I had the opportunity to review reports a longitudinal study about the impact of COVID-19-related social isolation on the mental health of young adults in Lombardy Region in the North of Italy, one of the first global hotspots of the pandemic.
While the idea of a longitudinal assessment of mental health in youths would be interesting, there are several issues that significantly affect the quality of this article.
In particular, two of them are remarkably relevant the scientific soundness of this piece of research.
1. First and foremost, the design of the study is hampered by several elements of inconsistency. The aim of the study reportedly is to investigate the impact of COVID-19-related social isolation on emotional and behavioural outcomes among health young adults comparing baseline and follow-up data, but:
1a. baseline data were collected, on average, more than a year and a half (81.8 ± 43.2 weeks) before the first day of lockdown. Several things may have changed in the participants’ lives that are not related to the pandemic and lockdowns, possibly representing significant confounders;
1b. for similar reasons, the SES of participants, which was not readdressed, may have changed, considering that this population has an age range that is known to be in a dynamic phase of life and change (e.g. moving from school to work);
1c. which questionnaire was administered to people who were 15-17 y.o. at baseline (originally administered with the YSR questionnaire) but then “grew up” from baseline to follow-up? Almost two years had passed from baseline to follow-up and most of those people likely became 18 or older, fulfilling the criteria for the ASR questionnaire;
1d. at baseline the questionnaires were completed with a trained psychologist, while at the follow-up they were administered via an online platform. This should be acknowledge as a methodological limitation.
2. Moreover, albeit acknowledged among the limitations, the sample size is too small for this study to be relevant.
Further observations:
3. Considering the topic of this study, a geographical reference in the title is much needed.
4. In the Introduction section, the insurgence of lockdown measures is described in detail. However, the Authors should as well describe how and when such lockdown measures were progressively dismissed so that the reader can understand what condition the participants were in when they were tested at follow-up.
5. Related to the previous point, The cut-off for low vs. high time spent in social isolation was chosen as 14 weeks. Was this an arbitrary choice? What were the restrictions at that time in Italy?
6. Methods, section LL 149-150: continuous variables are analysed using the median and the interquartile (IQR) range. Is it because the distribution is not normal? If so, the Authors should explicit it. Moreover, subsequently, Student t-tests with Welch’s correction to examine differences in demographic characteristics across the participants is not the correct test. Indeed, Mood's median test or the Wilcoxon-Mann-Whitney U test should be used.
7. The reporting of results in the relevant section of the manuscript is poor. More information regarding other subscales (now only in the supplement) should be reported.
8. In the Results section, subsection 3.3, the Authors state that “Spending a longer amount of time in social isolation (less than 14 compared to 14 weeks or more) was marginally associated with an increase of 1.73 points in rule-breaking behaviors”. However, p was 0.053, therefore, no significant association existed. This should be corrected.
9. In subsection 3.4, the p-value should be reported.
10. In Table 1, there is an error in the percentages of males at baseline.
11. The Discussion section is brief and superficial. Some aspects are not explored at all.
11a. In the Introduction, the Authors declare that their aim is “to quantify the impact of social isolation on young adult mental health to inform future interventions to minimize or eliminate the consequences of social isolation experience in healthy young adults”. However, the Discussion almost completely misses to provide suggestions on possible practical implications of the findings. What could have been done to address mental health issues in youths amid the COVID-19 pandemic? What can we learn for future times? Indeed, for example, evidence supports the utility of digital interventions in promoting the mental health of young people such as university students (Riboldi et al., 2022, https://doi.org/10.1016/j.rpsm.2022.04.005).
11b. There is no discussion at all about domains other than behavioural outcomes.
Reviewer 3 Report
Line 46 - remove the extra tab
Line 50 - had a tab
Line 52 - had a tab
Suggestion: check all the tabs in the paper
Line 64 - refer to what mental health problems they are talking
Line 102 - explain how adolescents got parental consent
Line 107 to 122 - what the questionnaire adapted to the Italian population. It needs to be explained
Line 148 - Since I’m not an expert in statists I will not comment on that, but it appears that the tests performed were the correct ones.
but more information on how the missing data and misses to follow-ups were treated is needed
244 - more explanation on the influenced bias is needed
Round 2
Reviewer 2 Report
I thank the Authors for having taken into consideration part of my suggestions. However, the Authors failed to satisfactorily address many of my points.
* Previous point #4: the Introduction now describes how lockdown measures were progressively dismissed, but timepoints are needed as well to understand the condition the participants were in when they were tested at follow-up. I suggest that the Authors are more specific in this regard.
* Previous point #7: in their reply, the Authors state that “Some additional information has been added”. However, no substantial changes were made to the Results section. I suggest that the Authors provide more details.
* Previous point #8: the phrase “marginally significant” does not mean anything. Once again, I strongly suggest that it is changed.
* Previous point #11: the Discussion was not revised.
o First, some statements about domains other than behavioural outcomes should be added.
o Second, suggestions on practical implications should be made. Digital interventions for youths should be mentioned (Riboldi et al., 2022, https://doi.org/10.1016/j.rpsm.2022.04.005).
Please refer to my previous review report for a more detailed explanation of my observations and suggestions.
Round 3
Reviewer 2 Report
The paper has improved and is now suitable for publication.